# Enhancing Transdermal Delivery: Investigating the Impact of Permeation Promoters on Ibuprofen Release and Penetration from Medical Patches—In Vitro Research

**DOI:** 10.3390/ijms242115632

**Published:** 2023-10-26

**Authors:** Paulina Bednarczyk, Anna Nowak, Wiktoria Duchnik, Łukasz Kucharski, Paula Ossowicz-Rupniewska

**Affiliations:** 1Department of Chemical Organic Technology and Polymeric Materials, Faculty of Chemical Technology and Engineering, West Pomeranian University of Technology in Szczecin, Piastów Ave. 42, 71-065 Szczecin, Poland; bednarczyk.pb@gmail.com; 2Department of Cosmetic and Pharmaceutical Chemistry, Pomeranian Medical University in Szczecin, Powstańców Wielkopolskich Ave. 72, 70-111 Szczecin, Poland; anowak@pum.edu.pl (A.N.); wiktoria.duchnik@pum.edu.pl (W.D.); lukasz.kucharski@pum.edu.pl (Ł.K.)

**Keywords:** ibuprofen, enhancers, acrylic pressure-sensitive adhesives, transdermal patch, shear strength, adhesion, tack

## Abstract

This study investigated the impact of various enhancers on permeation through the skin and accumulation in the skin from acrylic pressure-sensitive adhesive-based drug-in-adhesives matrix-type transdermal patches. Eleven patches, each containing a 5% enhancer of permeation, encompassing compounds such as salicylic acid, menthol, urea, glycolic acid, allantoin, oleic acid, Tween 80, linolenic acid, camphor, N-dodecylcaprolactam, and glycerin, were developed. Ibuprofen (IBU) was the model active substance, a widely-used non-steroidal anti-inflammatory drug. The results were compared to patches without enhancers and commercial preparations. The study aimed to assess the effect of enhancers on IBU permeability. The adhesive properties of the patches were characterised, and active substance permeability was tested. The findings revealed that patches with 5% allantoin exhibited the highest IBU permeability, approximately 2.8 times greater than patches without enhancers after 24 h. These patches present a potential alternative to commercial preparations, highlighting the significant impact of enhancers on transdermal drug delivery efficiency.

## 1. Introduction

For decades, the administration of drugs through the skin was primarily limited to local delivery, especially in the case of dermatological medications [1,2,3,4]. In recent years, the administration of drugs through the skin was principally confined to local delivery, especially in the case of dermatological medicines. However, there has been growing recognition of the significance of skin applications for addressing deeper tissue inflammation, such as muscles and joints, as well as for administering drugs to achieve systemic effects. This broader action of a drug, which extends beyond the skin’s surface despite its topical application, is made possible by the absorption of therapeutic substances into deeper tissues or the bloodstream [1,2,3,5,6,7,8].

As known, the transdermal administration of a drug to achieve systemic effects offers several significant advantages and often surpasses oral or intravenous drug delivery in many situations. Among the numerous benefits of transdermal drug delivery, particular attention is drawn to the ability to bypass first-pass metabolism in the liver and eliminate unwanted effects on the gastrointestinal tract [9,10,11,12,13]. Additionally, it prevents potential degradation of the active substance in the digestive system and avoids the possibility of drug interactions with food and other orally administered medications. This administration route allows for achieving the desired therapeutic effect with lower absorbed doses. With the implementation of suitable modifications to enhance permeability through biological membranes, these doses can be significantly reduced. In this context, the rate of drug absorption depends on the rate of its release, and this property can be accordingly controlled based on the desired effect. Furthermore, transdermal administration reduces the frequency of dosing for drugs with a short biological half-life, which is especially important in the treatment of chronic conditions [14,15,16,17,18].

Transdermal patches are pharmaceutical dosage forms designed for the controlled and sustained release of drugs through the skin. Typically, drugs used in transdermal patches are in the form of a soluble or dispersed drug in a matrix, reservoir, or adhesive system. The adhesive matrix in the patch ensures that it adheres to the skin and remains in place during use. The matrix should be biocompatible, non-irritating, and capable of maintaining adhesion for the intended duration of wear. The patch should securely adhere to the skin without causing discomfort or irritation. It should also be flexible enough to conform to the application site [19,20,21,22,23].

In the context of this method of drug administration, a critical factor that restricts penetration is the skin barrier. This barrier diminishes penetration efficiency and restricts the absorption of compounds. This layer represents the most formidable challenge in the transport of active substances and is widely recognised as the primary impediment to molecular permeation. It primarily consists of lipid components such as ceramides, cholesterol, fatty acids, cholesterol esters, and trace amounts of phospholipids. Among the various drugs available for topical application, only a relatively small subset can passively traverse the skin barrier in quantities substantial enough to achieve a therapeutic effect [20,24,25,26].

Among the numerous established techniques for enhancing the permeability of active substances through the skin, the utilisation of permeation promoters stands out as a noteworthy approach. These promoters facilitate the passage of active compounds through the skin barrier, thereby expanding the potential for effective transdermal drug delivery [10,27,28].

Penetration enhancers, also known as sorption promoters, infiltrate the *stratum corneum* and primarily enhance its permeability by disturbing the intercellular lipid arrangement [1,2,3,29]. An ideal sorption promoter should swiftly penetrate the skin’s *stratum corneum*, temporarily accumulate there, induce reversible changes in *stratum corneum* permeability, and be pharmacologically inactive, non-toxic, non-irritating, and non-sensitising. Furthermore, it should not engage in adverse interactions with other formulation components and should be organoleptically acceptable—odourless and colourless [10,28].

A crucial characteristic of a safe sorption promoter is its reversible action, allowing the restoration of the *stratum corneum* barrier after the topical application of the product. The return to full *stratum corneum* barrier function naturally occurs as the sorption promoter is eliminated from this layer. Typically, sorption promoters, such as other substances applied to the skin, eventually diffuse into the epidermis and the systemic circulation. The elimination of volatile substances (such as ethanol or terpenes) can also occur through vaporisation, which is advantageous as it reduces the risk of sorption promoter absorption into the bloodstream. The most commonly used sorption promoters include alcohols (ethanol), glycols (propylene glycol), unsaturated fatty acids (oleic acid), and terpenes (menthol). Infrequently, experimental studies may utilise azone, sulfolane (dimethyl sulfolane), surfactants, and other compounds from the abovementioned groups. Water, an unusual sorption promoter, can significantly alter *stratum corneum* permeability through substantial hydration [10,28,30].

The mechanism of action of penetration enhancers mainly depends on their polarity. Depending on their physicochemical properties, these compounds, when positioned within the *stratum corneum*, can disrupt the ordered arrangement of intercellular lipids in the *stratum corneum*, liquify or dissolve intercellular lipids, or alter the hydration of polar lipid groups. Notably, sorption promoters enhance the solubility of therapeutic substances in the *stratum corneum*, thereby increasing their partition coefficient between the *stratum corneum* and the underlying substrate [10,28,30,31].

The field of penetration enhancers for transdermal drug delivery is a dynamic and emerging area of research. As the need for non-invasive drug delivery methods continues to increase, understanding the mechanisms and safety and efficacy profiles of these enhancers becomes increasingly important. In this research work, we focused on showing the effect of different enhancers on the penetration of ibuprofen, an acrylic drug-in-adhesive matrix-type patch and their behaviour in a pressure-sensitive adhesive (PSA) matrix with self-adhesive properties. Eleven patches were developed, each containing a 5% enhancer of permeation. These enhancers were selected based on their potential to disrupt the stratum corneum and enhance permeability, and they encompassed compounds such as salicylic acid, menthol, urea, glycolic acid, allantoin, oleic acid, Tween 80, linolenic acid, camphor, N-dodecylcaprolactam, and glycerin. The choice of these specific penetration enhancers was driven by their varying physicochemical properties and known effects on skin permeation. These enhancers were chosen to represent a range of chemical classes, including alcohols, fatty acids, and other compounds, that have shown potential as permeation enhancers in previous research. Their selection aimed to provide a comprehensive assessment of the impact of different enhancers on the transdermal delivery of ibuprofen.

Furthermore, release from the patch, skin permeability, and accumulation of the active substance in the skin were investigated and compared, with the results obtained from the transdermal patch without enhancer and commercial preparation.

## 2. Results and Discussion

### 2.1. Examination of Self-Adhesive Characteristics in a Transdermal Patch with Diverse Enhancers

The pressure-sensitive adhesive (PSA) utilised in this study comprised a proprietary acrylate copolymer, incorporating components such as 2-ethylhexyl acrylate, hydroxyethyl acrylate, glycidyl methacrylate, and vinyl acetate. This acrylic PSA underwent thermal crosslinking to establish a crosslinked polymer matrix. Since both ibuprofen and enhancers were present in the PSA during the crosslinking procedure, it becomes imperative to assess their impact on the crosslinking process and the subsequent self-adhesive attributes (as indicated in Table 1). In order to determine the effect of enhancers on the self-adhesive properties, patches were first made without ibuprofen and then a series with the addition of enhancers and the drug simultaneously in the patch.

Therefore, the prepared patches had different coat weights, which in the case of the series without the drug were in the range of 29–39 g/m^2^ and, respectively, in the case of the series with ibuprofen were in the range of 26–40 g/m^2^. All patches were crosslinked under the same conditions. However, the amount of solvents used to dissolve the active substances differed, influencing the solid weight content determined via gravimetry.

Shear strength (shear adhesion) reveals the resistance of a transdermal patch to tangential stresses and, therefore, the cohesion of the adhesive matrix [33]. Studies have shown that the addition of permeation promoters only at a concentration of 2 wt. did not reduce the cohesive forces of the adhesive matrix. The exception was the use of glycolic acid and alpha-linolenic acid. In this case, the cohesion significantly decreased. Furthermore, the significant changes are noticeable when ibuprofen is added. As we described in our previous publication [32], the addition of the active substance significantly reduces the cohesion of the adhesive matrix in relation to the corresponding samples without ibuprofen. However, if we compare individual patches with IBU and various penetration promoters to the reference patch (patch with IBU, but without the penetration promoter—TP-IBU), it can be seen that some active substances can improve shear strength (>10 min). This type of property is noticed when using salicylic acid (26 min), camphor (13 min), urea (27 min), and allantoin (24 min).

Effective adhesion is a key feature of any transdermal patch, as the amount of active substance delivered to the skin is dependent on the patch’s surface area. Therefore, the partially detached surface may decrease the quantity of ibuprofen that penetrates the skin. This is especially crucial when the patch must be worn for an extended period of time, such as seven days. To resolve this issue, the patches must firmly adhere to the epidermis for the duration of this period. Therefore, the impact of the patches with various permeation promoters in an adhesive matrix on the adhesion of the resulting patch was assessed. It turns out that most of the permeation promoters used that were incorporated into the adhesive matrix acted as adhesion promoters. A significant improvement in this property was noted in comparing the results of the adhesion of individual patches to the reference sample. This is probably due to the presence of functional groups in the chemical structure of the permeation promoters used, such as hydroxyl or acid groups. The only exception was the use of n-dodecylcaprolactam and tween 80. In this case, slightly lower results were obtained, but the adhesion of these samples was still considered good. Similar results were obtained with samples containing ibuprofen. However, lower adhesion was obtained in the samples with urea and n-dodecylcaprolactam in this case.

The tack test assessed the effectiveness of transdermal patch adhesion by measuring the force of debonding on applying light pressure for a short duration of time. The tack of the resulting patches was improved using glycerol, salicylic acid, L-menthol, and camphor. In the remaining cases, deterioration of this property was obtained. On the other hand, in the case of the ibuprofen series, all samples had a lower tack compared to the reference sample (TP-IBU, 13.50 N).

### 2.2. Microscopy and Stability Evaluation of Acrylate-Based Transdermal Patches with Various Enhancers

In this study, we conducted a short-term stability assessment for the optimised formulation after 7 days and 3 months to gauge its quality and estimate the potential shelf-life of the resulting patches. The samples were shielded with siliconized film to simulate storage conditions and maintained at constant temperatures. Comparative analysis was also carried out on patches without protective foil. Furthermore, in the case of patches not protected with siliconized foil, a microscopic examination was carried out, especially after 7 days corresponding to the period of use of the patch on the skin. We evaluated all samples for any colour alterations through organoleptic analysis and assessed crystallinity using microscopy. Our investigation considered the chemical structure of the enhancers used, including their categorisation, polarity, crystallisation tendencies, and diffusivity (Table 2). The research findings underscore the significant influence of these variables on drug permeability through the skin and the functional attributes of the patches.

The enhancers used in this research can be categorised into various classes, including organic acids (e.g., salicylic acid, glycolic acid), alcohols (e.g., L-menthol, glycerin), amides (e.g., allantoin), fatty acids (e.g., oleic acid, linolenic acid), surfactants (e.g., Tween 80), and more. This categorisation helps in understanding their chemical properties and functions. These compounds are with different polarities. Polarity influences a compound’s ability to interact with other polar or nonpolar molecules, impacting factors such as solubility and diffusion through biological membranes. Compounds such as glycolic acid and salicylic acid are polar due to their hydroxyl (OH) functional groups, while compounds such as camphor are relatively nonpolar. Crystallisation tendencies vary among used enhancers. For instance, salicylic and glycolic acids easily crystallise due to their chemical structure and polarity. On the other hand, some compounds such as L-menthol and camphor can form crystalline structures under specific conditions, especially at lower temperatures. Crystallisation can affect the stability and appearance of transdermal patches. Compounds such as salicylic acid, glycolic acid, and urea, being relatively small and polar, may have higher diffusivity through skin layers. Others, such as N-dodecylcaprolactam, a larger and less polar compound, may exhibit lower diffusivity. The diffusivity of these compounds through the skin is a critical factor in their transdermal delivery efficacy. These characteristics play a crucial role in the formulation and performance of transdermal patches, influencing drug release, permeation, and stability. Additionally, they affect the interaction of these compounds with the skin’s barrier and their ability to achieve the desired therapeutic effect.

Table 3 displays the resulting transdermal patches. As a result of their observation using organoleptic methods, in the case of most of the tested samples, no significant changes were found during the seasoning time. Overall, the transdermal patches demonstrated stability, with no observed crystallisation over time. An exception was the menthol-containing sample, where, after 3 months, crystallised enhancer substance regions were noted. In the case of allantoin, sparingly soluble particles within the adhesive matrix were observed from the outset.

Table 4 shows patches that were stored without protective foil, allowing air exposure. Consequently, significant active substance crystallisation occurred in five samples, specifically those containing glycerin (GL), oleic acid (OA), menthol (Menth), allantoin (All), and N-dodecylcaprolactam (LC).

### 2.3. Microspectroscopy Analysis

Interactions between the adhesive matrix and the introduced active substances were examined using infrared spectroscopy (FTIR). Figure 1 shows the FTIR spectra of patch samples containing ibuprofen and various permeation promoters. All spectra show peaks corresponding to the characteristics of the acrylate adhesive matrix, i.e., strong vibrations are present at 1730 cm^−1^ contributed by -C=O of acrylate groups, and peaks that simultaneously occur in the adhesive matrix and the structure of ibuprofen, i.e., corresponding to the hydroxyl groups and hydrogen bonds in the range of 3000–2800 cm^−1^, and corresponding to the bonds stretching vibrations between C-O atom characteristics for ether (Ar-OR), ester (RCOOR’), and carboxylic acid (RCO-OH) groups, visible as bands at wavenumber 1235, 1163, 1093, 1015, and 966 cm^− 1^, while bands at wavenumber 1458 cm^−1^ correspond to the C-C bonds in the aromatic ring. The mentioned peaks are observed for all tested patches, and no shifts or significant changes in intensity or shape are observed in this case. On the other hand, the presence of a distinct peak at 1705 cm^−1^, corresponding to the bonds stretching vibrations between C=O atoms characteristic for carboxylic groups (RCO-OH), is characteristic for all patches containing ibuprofen in comparison to the patch without ibuprofen (TP). It was also observed that the addition of various penetration promoters affects the intensity of the bands at 1340, 1015, 870, and 720 cm^−1^ corresponding to ether, ester, or carboxyl groups, which is related to the presence of these characteristic groups in the structures of the additives used. Additionally, Appendix A shows FT-IR spectra of the pure form of the permeation promoters.

The IR method equipped with a microscope was particularly helpful in identifying the crystallised substance in patches stored without the protection of the adhesive layer. This method allows for viewing the transdermal patch’s specified measurement site and simultaneously collecting the infrared spectrum. The FTIR spectra of patch samples containing ibuprofen and various permeation promoters made in the areas of crystallisation are presented in Figure 2, along with the reference sample, which was ibuprofen. Samples were also analysed for the presence of various permeation promoters (spectra of the pure form of the permeation promoters used are provided in Appendix A). The spectra of the crystallised substance are almost identical, which indicates that it is probably ibuprofen. This may be confirmed by the presence of peaks characteristic of ibuprofen at 1705 cm^−1^, corresponding to the bonds stretching vibrations between C=O atoms characteristic for carboxylic groups (RCO-OH), or characteristic of ibuprofen peaks at 695, 670, 635, and 590 cm^−1^. Moreover, shifts of peaks 1458, 1235, and 870 cm^−1^ to 1461, 1230, and 865 cm^−1^, respectively, were observed, which exactly occur in these places in the pure form of ibuprofen. The presence of additional peaks, e.g., at 1730 cm^−1^ and 1163 cm^−1^, which are not present in the IBU spectrum, indicates that information has also been collected regarding the adhesive matrix surrounding the crystallised substance.

### 2.4. Thermal Properties

The thermal stability assessment of the produced patches was conducted through thermogravimetric analysis, and several key parameters were determined, including the onset decomposition temperature, temperature corresponding to 50% weight loss (derived from TG curves), and maximum decomposition temperatures (determined from DTG curves). These properties are compiled in Table 5 and Appendix A (available in the Appendix A). Notably, the adhesive layer of patches lacking the addition of the active substance exhibited greater stability compared to those containing ibuprofen or ibuprofen with enhancers, consistent with our prior research findings [32]. The alteration in the adhesive layer’s stability is attributed to the inclusion of an active substance with lower stability than that of the adhesive. Specifically, the onset of degradation for TP is approximately 318 °C, whereas, for medical patches, this value ranged from 159.0 to 178.0, lower for TP-IBU-SA and TP-IBU-All, respectively. The temperatures corresponding to a 50% weight loss of adhesives varied from 336.3 (for TP-IBU-GA) to 360.0 °C (for TP). Importantly, these values remained consistent regardless of whether the adhesive contained an active substance or active substance with an enhancer. The maximum decomposition temperatures spanned from 331.9 °C for TP-IBU-LC to 383.1 °C for TP-IBU-All. In the majority of cases, the maximum decomposition temperatures exceeded the temperatures, corresponding to 50% weight loss. The table also shows the glass transition temperature recorded in the second heating cycle determined on the basis of DSC thermograms (a summary of DSC thermograms is provided in Appendix A). The glass transition temperature of the obtained patches with ibuprofen and various penetration promoters ranged from −50.62 to −55.68 °C, so it was very close to the temperature of the patch only containing ibuprofen, which probably translates into the lack of chemical interactions with the adhesive matrix.

### 2.5. Permeability, Release, and Accumulation in Skin Studies

The cumulative mass of ibuprofen in the acceptor fluid at all time points is depicted in Figure 3 and Appendix A. In contrast, the IBU content in the acceptor fluid collected during the 24-h permeation is summarised in Table 6. After 24 h of permeation, the cumulative mass of IBU followed this order: TP-IBU-OA > TP-IBU-All > TP-IBU-GA > TP-IBU-LC > TP-IBU-Urea > TP-IBU-GL > TP-IBU-T80 > TP-IBU-Menth > TP-IBU-Cam > TP-IBU-LA > TP-IBU-SA > CP > TP-IBU. Among the tested patches, TP-IBU-OA exhibited the highest penetration of ibuprofen, with a cumulative permeation of 163.306 ± 24.418 µg/cm^2^ (Table 6 and Figure 3).

Permeability parameters were assessed, including flux (J_SS_, µg/cm^2^∙h), apparent permeability coefficient (K_P_∙10^3^, cm/h), and the percentage of drug that penetrated after 24 h (Q%_24 h_). The results are summarised in Table 6, revealing noticeable variations in permeation among patches containing different enhancers. The lowest permeation rate was observed for TP-IBU (68.386 µg/cm^2^∙h), while TP-IBU-OA exhibited the highest permeation rate at 163.306 ± 24.418 µg/cm^2^∙h. Notably, the use of enhancers resulted in higher levels of active substance permeation after 24 h. Steady-state flux values varied based on the release profile, with TP-IBU-All demonstrating the most increased flux. The permeability coefficient, a quantitative indicator of the rate at which molecules penetrate the skin, was also determined. This parameter is influenced by various factors related to the drug, skin barrier, and their interactions. For the tested patches, permeability coefficient values ranged from 4.71∙10^3^ cm/h for TP-IBU-LC to 14.89∙10^3^ cm/h for TP-IBU-All. The percentage of ibuprofen that permeated through the skin was also calculated. TP-IBU-OA exhibited the highest permeation percentage. These findings indicate that the enhancers tested in this study may indeed enhance skin permeability.

The cluster analysis graph illustrates the cumulative mass of IBU measured throughout the entire 24-h permeation period (see Appendix A). Within this graph, two clearly distinguishable groups of patches, denoted by green circles (indicating lower penetration) and red circles (indicating higher penetration), are evident. Overall, the incorporation of permeation enhancers enhances the permeability of ibuprofen. It is evident that ibuprofen permeates more effectively from patches with penetration enhancers compared to patches without enhancers and a commercial product (see Appendix A).

The permeation rate, which gauges the therapeutic effectiveness achieved, was also assessed. Figure 4 depicts the permeation rate observed for each time interval. In general, the highest permeation rate for all patches was observed within the initial half-hour of measurement. It is evident that all enhancers employed in the study contribute to an increased permeation rate of ibuprofen.

The liberation of the active compound from the medicinal patch was also assessed. As depicted in Figure 5, the most substantial quantities of the active pharmaceutical ingredient (API) are liberated during the initial 3 h, following which the API release experiences a minor reduction. Consequently, the acquired patches do not impose constraints on the discharge of active components. These findings imply that medical patches designed for pain relief can potentially ensure heightened efficacy compared to commercially accessible products.

Figure 6 shows the mass of IBU that accumulated in pigskin after 24 h of penetration, expressed in μg IBU/g of skin. All of the patches containing promoters used caused the accumulation of the drug in the skin. The highest statistically significant accumulation in the skin was for IBU released from TP-IBU-LA (103.590 ± 8.472 μg/g of skin) and TP-IBU-GL (101.427± 8.339 μg/g of skin)—Figure 6.

The cluster analysis graph (see Appendix A) displays the cumulative IBU mass released over 24 h, revealing two distinct groups of patches: green circles indicating lower release and red circles indicating higher release. Overall, the inclusion of permeation enhancers significantly improves ibuprofen’s permeability. Notably, patches with penetration enhancers demonstrate more effective ibuprofen release compared to those without enhancers and a commercial product (see Appendix A).

## 3. Materials and Methods

### 3.1. Materials

The drug-in-adhesives matrix-type transdermal patch was obtained using a commercial polyacrylate adhesive—DURO-TAK 378-2054. This adhesive, comprising an acrylate copolymer, was formulated in a solvent blend containing propan-2-ol (10–20%), ethyl acetate (10–20%), n-heptane (1–5%), petroleum (1–5%), methylcyclohexene (1–5%), and toluene (1–3%) and characterised by viscosity: 1.46 Pa·s and SWC: 49.7%. Ibuprofen (99%) (IBU), as an active substance, was obtained from Sigma Aldrich (Steinheim am Albuch, Germany). Additionally, various penetration promoters were used, including salicylic acid (SA, ≥99.0%, Sigma Aldrich), L-menthol (Menth, 99%, Sigma Aldrich), urea (≥99.9%, Chempur, Piekary Śląskie, Poland), glycolic acid (GA, ≥98%, AmBeed), allantoin (All, ≥98%, Sigma Aldrich), oleic acid (OA, 100%, Warchem sp. z o.o.), Tween 80 (Polysorbate, polyethylene sorbitol ester, T80, Croda, Snaith, England, UK), linolenic acid (LA, ≥99%, Sigma Aldrich), camphor (Cam, ≥96%, Thermo Scientific, Across Organic, Waltham, MA, USA), N-dodecylcaprolactam (LC, 97%, AmBeed), and glycerin (GL, ≥99.5%, Chempur).

Additional reagents employed in the permeation tests included PBS buffer at pH 7.4 (sourced from Merck, Darmstadt, Germany), high-purity orthophosphoric acid (98%) acquired from Chempur, HPLC gradient-grade acetonitrile (≥99.9%), and methanol (99.9%) obtained from Sigma-Aldrich.

### 3.2. Preparation of Transdermal Patches

The adhesive matrix for the transdermal patches contained commercial acrylate copolymer (DURO-TAK 378-2054; DT54; viscosity: 1.46 Pa·s; SWC: 49.7%). First, a patch consisting only of the adhesive matrix was prepared, i.e., without the addition of IBU and penetration promoters (TP). Then, a patch with IBU (TP-IBU) and patches containing both IBU and various permeation promoters (2% by weight in relation to the adhesive). Solid substances were previously dissolved in a suitable solvent and then incorporated into the adhesive. In turn, the weight ratio of the adhesive matrix to active substance was calculated based on the adhesive characteristics, i.e., solids content, the basis weight depends on the applied thickness of the adhesive film, and the characteristics of the active substance, i.e., the molar mass and the initial assumption regarding the content of active substances in commercial products, i.e., 200 mg of the active substance (ibuprofen) for the surface of the adhesive film equal to 140 cm^2^. The adhesive compositions in this series were prepared by dissolving the active substance in ethyl acetate and then adding the mixture to the adhesive matrix containing the permeation promoter. Then, the adhesive compositions were coated (250 µm) on a polyester film. The obtained polymer layers were thermally crosslinked in the next stage for 15 min at 110 °C. The resulting adhesive film layer was covered with siliconized release paper. Table 7 shows the appropriate weights with which to prepare the formulation.

### 3.3. Properties of Transdermal Patches

Viscosity measurements of the resulting adhesive compositions were conducted using a Bohlin Visco 88 viscometer (Malvern Panalytical, Malvern, UK). The measurements were carried out at 20 °C with a C14 geometry, operating at 20 rpm.

The solid weight content (SWC) was determined following ISO 3251 guidelines, involving heating at 140 °C for 30 min, employing a moisture analyser (Radwag MAX 60/NP).

To determine the coat weight of the crosslinked adhesive films after solvent evaporation, a circular punch 1009 with a 10 cm^2^ area (Karl Schröder KG, Weinheim, Germany) was used.

Thermal stability was assessed through thermogravimetric analysis using a thermomicrobalance TG 209 F1 Libra from Netzsch. Approximately, 5-mg samples were heated at a rate of 10 °C/min in an oxidative atmosphere (nitrogen as a protective gas at 10 mL/min and air at 25 mL/min) over a temperature range of 25 to 1000 °C. The onset decomposition temperature was determined based on the intersection of TG curve tangents, while the temperatures corresponding to the most rapid sample weight loss were identified from the first derivative of the TG curve (DTG curve).

For the determination of the adhesive’s glass transition temperature, differential scanning calorimetry (DSC) analysis was carried out using a Q-100 differential calorimeter (TA Instruments, New Castle, DE, USA, 2004). Samples were subjected to a heating cycle ranging from −90 °C to +190 °C at a heating rate of 10 °C/min.

### 3.4. Self-Adhesive Properties of Transdermal Patches

Various self-adhesive properties were investigated, including tack, adhesion, and cohesion at different temperatures for the coated, crosslinked adhesive. These evaluations adhered to international standards from AFERA and FINAT. Furthermore, shear strength testing was conducted following FINAT FTM 8 guidelines, adhesion was assessed per AFERA 4001 standards, and tack was evaluated in accordance with AFERA 4015 guidelines. All tests were performed using a Zwick/Roell Z-25 testing machine, with a detailed procedure available in our previous article [34].

### 3.5. Microscopic Examination and Stability Assessment of Transdermal Patches

The prepared acrylate patches were stored under constant conditions at 20 °C with 50% humidity. An optical microscope (Delta Optical) equipped with an MC500-W3 5MP camera was employed to detect crystallisation. The camera, attached to the microscope, captured images at a magnification of 10×. The presence of drug crystals was monitored for seven days in patches without siliconized foil protection, mimicking the period of patch used on the skin, and for three months in patches shielded with siliconized foil to simulate storage conditions.

### 3.6. Infrared Microspectroscopy

Each patch was analysed using a Nicolet iS5 IR spectrophotometer (Thermo Scientific, Waltham, MA, USA) covering 500 to 4000 cm^−1^. An additional SurveyIR™ Infrared Microspectroscopy accessory (Thermo Scientific, Czitek, LLC, Waltham, MA, USA) was utilised. Visual images were generated using a high-resolution 5MP CMOS colour video camera with a maximum resolution of 2592 × 1944 and a field of view measuring 1900 µm. The eSpot software (version 1.0.57) facilitated image display, manipulation, capture, and documentation, providing an interface for selecting sampling and illumination modes. This approach enabled the simultaneous viewing and collection of infrared (IR) spectra. Samples were placed in the spectrometer’s sample compartment for microsampling, and observations were conducted in transmission mode. Following the selection of the analysis site, FTIR spectra were recorded by collecting data in ATR mode.

### 3.7. In Vitro Skin Permeation, Release, and Accumulation Studies

For the permeation experiments, we employed Franz diffusion cells (Phoenix DB-6, ABL&E-JASCO, Wien, Austria) with a diffusion area of 1 cm^2^. The acceptor chamber, equipped with a magnetic stirring bar, was filled with 10 mL of pH 7.4 PBS solution. To maintain constant conditions, a temperature of 37.0 ± 0.5 °C and a consistent stirring speed were employed. Porcine skin was chosen for the experiment due to its similar permeability to human skin. Fresh abdominal porcine skin was acquired from a local slaughterhouse, washed in PBS buffer pH 7.4, and dermatomed to a thickness of 0.5 mm. These skin samples were stored in aluminium foil in a freezer at −20 °C for three months to preserve their skin barrier properties. Before use, the skin samples were slowly thawed at room temperature for 30 min and hydrated with PBS pH 7.4. Only skin samples with impedance >3 kΩ, akin to the electrical resistance of human skin, were utilised [35,36].

The patches were applied to the skin, and as previously described, the experiment ran for 24 h, with samples collected at specified intervals. Release tests followed the same procedure as the permeation tests but with a membrane. A dialysis tubing cellulose membrane (D 9777-100FT, Sigma Aldrich, Steinheim am Albuch, Germany) covered the diffusion areas, and the prepared patch was placed over the membrane. The acceptor cell contained 10 mL of pH 7.4 PBS, maintained at 37 °C, and the experiment spanned 24 h. It is worth noting that the solubility of ibuprofen in a PBS solution with pH 7.4 is 0.432 ± 0.001 g/dm^3^ [37]. Samples were taken at intervals of 10, 20, 30, 40, 50, 60, 70, 80, 100, 120 min, and 24 h, with 0.4 mL of acceptor fluid aliquots withdrawn and replaced with fresh buffer of the same pH. HPLC was used to measure compound concentrations in the acceptor fluid.

The accumulation of the tested substance in the skin post penetration was determined using established methods [32,37,38,39,40]. The supernatant was collected and analysed using HPLC. Accumulation of IBU in the skin was calculated as the ratio of the drug remaining in the skin to the mass of the skin sample, expressed as μg/g. The liquid chromatography system (Knauer, Berlin, Germany) assessed IBU and its derivatives in the acceptor fluid during permeation and release tests, as well as accumulation in the skin.

The HPLC system comprised a model 2600 UV detector, a Smartline model 1050 pump, and a Smartline model 3950 autosampler with ClarityChrom 2009 software (Knauer, Berlin, Germany). The detector operated at 264 nm. A chromatographic column measuring 125 × 4 mm filled with Hyperisil ODS (C18), with a particle size of 5 µm, was employed. The mobile phase consisted of 0.02 M potassium dihydrogen phosphate–acetonitrile (60/40 *v*/*v*) with a flow rate of 1 mL min^−1^. The column temperature was maintained at 25 °C, and the injection volume was 20 μL.

The IBU and its derivative concentrations in the acceptor phase were determined by HPLC. Cumulative mass (µg/cm^2^) was calculated based on this concentration. Flux (µg/cm^2^·h) through pigskin into the acceptor fluid was determined as the slope of the cumulative mass plot against time.

### 3.8. Statistical Analysis

The results are expressed as the mean ± standard deviation (SD). A one-way analysis of variance (ANOVA) was conducted for statistical analysis. Tukey’s test (α < 0.05) was applied to evaluate the significance of differences between individual groups in cumulative mass. A cluster analysis was also performed to identify similarities among all tested patches, considering all time points for acceptor fluid collection. This analysis grouped compounds with similar permeation patterns. Statistical computations were performed using Statistica 13 PL software (StatSoft, Kraków, Poland).

## 4. Conclusions

This study underscores the complex interplay between enhancers, ibuprofen, and adhesive properties in transdermal patches. The choice of enhancer can have a significant impact on the adhesive characteristics of the patch, with some enhancers improving shear strength and adhesion. In contrast, others may differently affect these properties. Understanding these interactions is crucial for the development of effective transdermal drug delivery systems. The research demonstrates that the inclusion of various enhancers in acrylic pressure-sensitive adhesive-based transdermal patches can significantly impact the permeation of ibuprofen through the skin and its accumulation in the skin. These findings have implications for the design and formulation of transdermal patches for enhanced drug delivery and therapeutic efficacy. Enhancers such as oleic acid (OA) and allantoin (All) showed the highest cumulative permeation of ibuprofen, indicating their effectiveness in enhancing drug delivery. TP-IBU-OA exhibited the highest permeation rate, while other enhancers also contributed to increased drug permeation compared to patches without enhancers. The research revealed that the choice of permeation enhancer significantly influences the permeation of ibuprofen through the skin. Notably, TP-IBU-OA (containing oleic acid) exhibited the highest cumulative permeation of ibuprofen, with a value of 163.306 ± 24.418 µg/cm^2^, highlighting the remarkable efficacy of this enhancer in facilitating drug delivery across the skin. The results suggest that the enhancers tested in the study have the potential to enhance skin permeability, which is crucial for transdermal drug delivery systems aiming to achieve therapeutic efficacy. The release profiles of ibuprofen from the patches showed that the majority of drug release occurs within the initial 3 h. Importantly, the patches did not impose constraints on the discharge of active components, indicating their ability to provide sustained drug release.

## Figures and Tables

**Figure 1 ijms-24-15632-f001:**
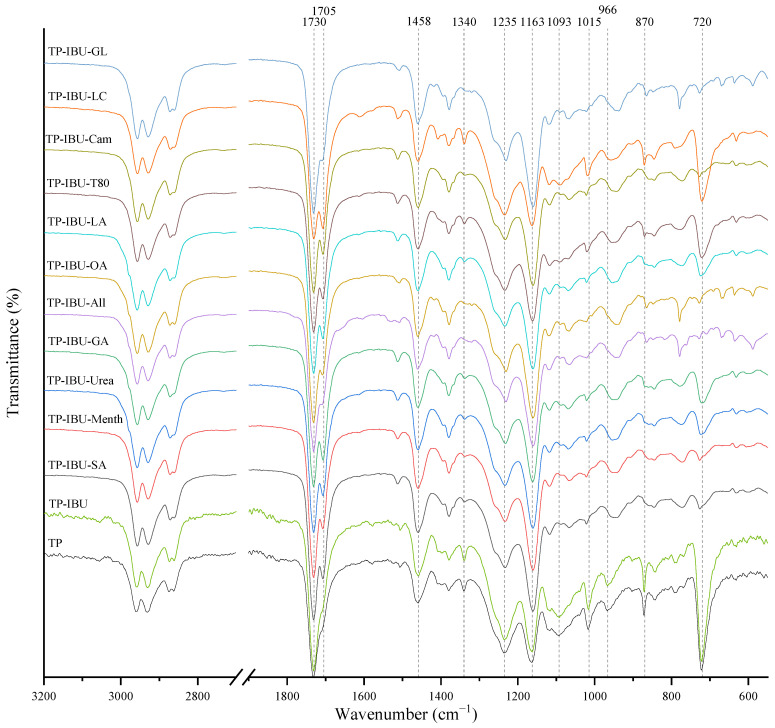
FTIR spectra of patch samples containing ibuprofen and various permeation promoters.

**Figure 2 ijms-24-15632-f002:**
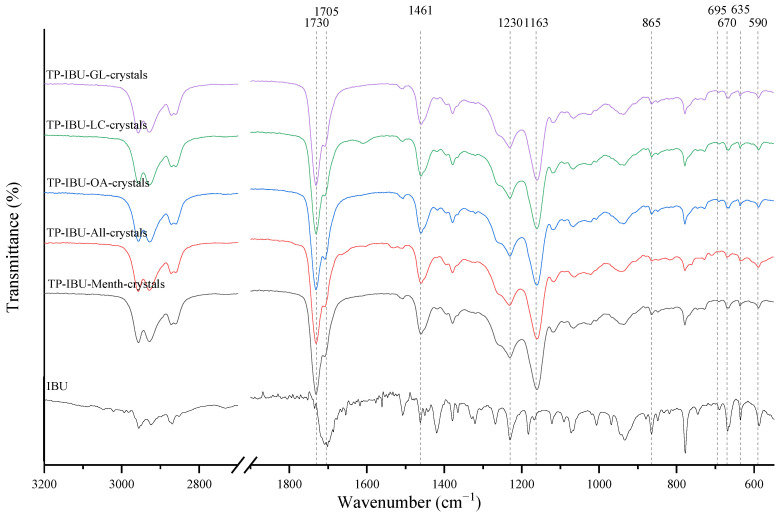
FTIR spectra of patch samples containing ibuprofen and various permeation promoters taken from the areas of crystallisation.

**Figure 3 ijms-24-15632-f003:**
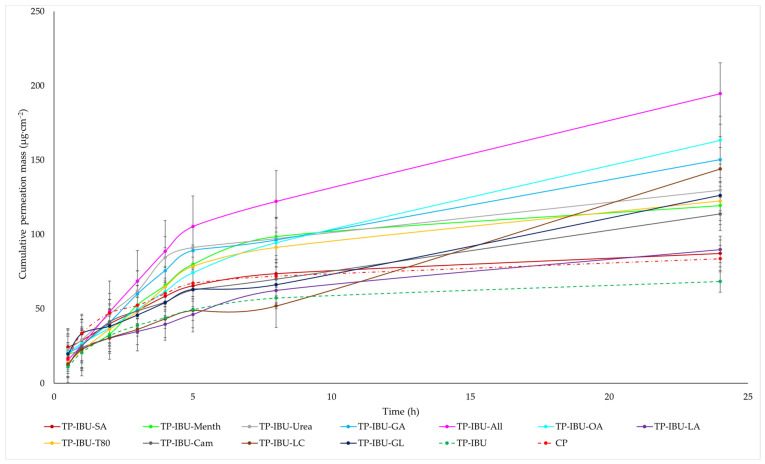
Time course of the cumulative mass of IBU during the 24-h permeation (Mean ± SD, *n* = 3).

**Figure 4 ijms-24-15632-f004:**
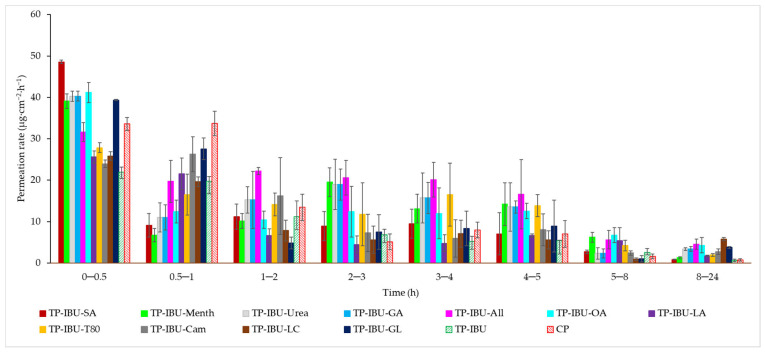
The permeation rate of IBU during the 24-h permeation; α = 0.05 (Mean ± SD, *n* = 3).

**Figure 5 ijms-24-15632-f005:**
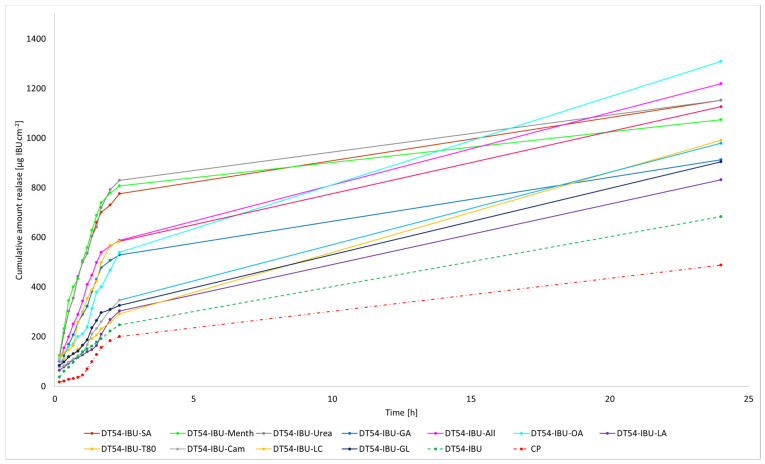
Time course of the cumulative mass of IBU during the 24-h release (Mean ± SD, *n* = 3).

**Figure 6 ijms-24-15632-f006:**
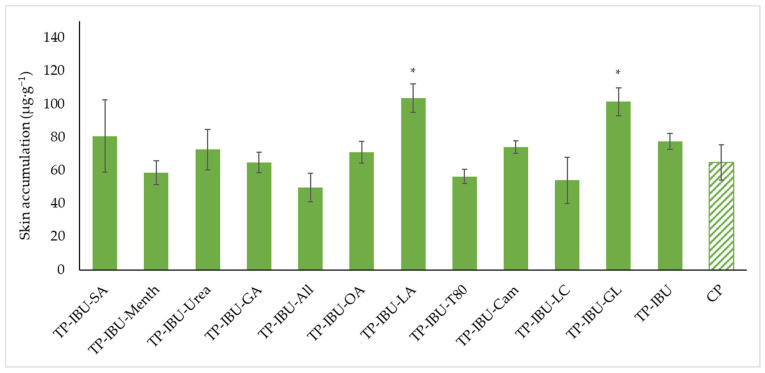
Accumulation in the skin of IBU after the 24-h penetration. Each point represents the mean ± SD (*n* = 3), α = 0.05, *—significantly higher accumulation in the skin versus the control (TP-IBU). The statistically significant difference was estimated using the ANOVA test (Tukey’s test).

**Table 1 ijms-24-15632-t001:** Self-adhesive properties of the transdermal patch containing various enhancers.

Sample Code	Coat Weight [g/m^2^]	SWC [%]	Shear Strength	Adhesion [N/25 mm]	Tack [N]
TP	32 [32]	98 [32]	>72 h [32]	13.60 [32]	14.00 [32]
TP-IBU	40 [32]	97 [32]	10 min/c.f. [32]	11.90/c.f. [32]	13.50 [32]
TP-SA	38	98	>72 h	21.50	16.58
TP-Menth	29	99	>72 h	16.84	16.27
TP-Urea	31	98	>72 h	15.03	10.25
TP-GA	37	99	7 min/c.f.	22.69	11.73
TP-All	32	100	>72 h	14.29	12.10
TP-OA	39	99	>72 h	15.78	7.69
TP-LA	35	99	12 h 7 min/c.f.	13.80	7.60
TP-T80	37	99	>72 h	10.75	11.57
TP-Cam	31	99	>72 h	20.71	16.35
TP-LC	33	99	>72 h	12.30	9.01
TP-GL	31	99	>72 h	23.33	13.82
TP-IBU-SA	34	94	26 min/c.f.	22.24	11.30
TP-IBU-Menth	40	92	9 min/c.f.	12.31	9.34
TP-IBU-Urea	31	93	27 min/c.f.	6.48	1.27
TP-IBU-GA	26	94	2 min/c.f.	11.96	12.27
TP-IBU-All	31	91	24 min/c.f.	13.24	11.24
TP-IBU-OA	33	93	10 min/c.f.	12.27	7.45
TP-IBU-LA	27	96	9 min/c.f.	18.94	10.27
TP-IBU-T80	38	93	7 min/c.f.	14.00	11.23/c.f.
TP-IBU-Cam	30	93	13 min/c.f.	14.55	11.01
TP-IBU-LC	37	94	1 min/c.f.	5.87	13.45/c.f.
TP-IBU-GL	26	94	2 min/c.f.	11.96	12.27

SWC—Solid weight content determined via gravimetry; TP—transdermal patches (containing adhesives); TP-IBU—TP with ibuprofen; TP-IBU-SA—TP with IBU and salicylic acid; TP-IBU-Menth—TP with IBU and menthol; TP-IBU-Urea—TP with IBU and urea; TP-IBU-GA—TP with IBU and glycolic acid; TP-IBU-All—TP with IBU and allantoin; TP-IBU-OA—TP with IBU and oleic acid; TP-IBU-LA—TP with IBU and linoleic acid; TP-IBU-T80—TP with IBU and Tween 80; TP-IBU-Cam—TP with IBU and camphor; TP-IBU-LC—TP with IBU and N-dodecylcaprolactam; TP-IBU-GL—TP with IBU and glycerine; c.f.—cohesive failure.

**Table 2 ijms-24-15632-t002:** Categorisation, polarity, crystallisation tendencies, and diffusivity of used enhancers.

Enhancer	Categorisation	Polarity	Crystallisation Tendencies	Diffusivity
SA	beta-hydroxy acid (BHA) and is commonly used in skincare products	moderately polar compound due to its hydroxyl group, making it soluble in both water and lipids	can crystallise under certain conditions, particularly when concentrated or exposed to lower temperatures	has reasonable diffusivity, allowing it to penetrate the skin to some extent
Menth	naturally occurring terpene alcohol	moderate polarity and its chemical structure includes a hydroxyl group	can crystallise at lower temperatures or when it becomes concentrated	has relatively good diffusivity, which contributes to its cooling sensation when topically applied
Urea	naturally occurring organic compound (diamide) and is often used in skincare for its moisturising properties	polar compound due to its amide functional group and exhibits good water solubility	tends to form crystals under certain conditions, especially when the solution becomes concentrated	has moderate diffusivity and can penetrate the skin to provide hydration
GA	alpha-hydroxy acid (AHA) and is used in skincare for its exfoliating properties	polar compound with good water solubility	may crystallise in concentrated solutions	has good diffusivity, which allows it to effectively penetrate the skin and exfoliate
All	chemical compound (acylureas) with anti-irritant and skin-soothing properties	moderately polar and exhibits good water solubility	can crystallise when concentrated	moderate diffusivity and can penetrate the skin to provide soothing effects
OA	monounsaturated omega-9 fatty acid	moderately polar due to its carboxylic acid group	can crystallise at lower temperatures	moderate diffusivity and can penetrate the skin
LA	polyunsaturated fatty acid	moderately polar due to its multiple double bonds	can crystallise under certain conditions	moderate diffusivity and can penetrate the skin
T80	non-ionic surfactant, often used as an emulsifier	Amphiphilic	is not known for crystallisation	enhances the dispersion of other compounds, potentially improving their penetration
Cam	terpene and is often used in topical products for its cooling sensation	moderately polar due to its functional groups	can crystallise at lower temperatures	has reasonable diffusivity, contributing to its sensory effects
LC	caprolactam compound	moderately polar due to its structure	can crystallise under certain conditions	diffusivity depends on its formulation
GL	sugar alcohol	highly polar and hygroscopic, with excellent water solubility	typically does not crystallise	has good diffusivity, and its hygroscopic nature can help maintain skin hydration

**Table 3 ijms-24-15632-t003:** Microscopic observation of patch samples containing ibuprofen and various permeation promoters during seasoning (patches protected with siliconized foil).

Sample Code	First Day of Observation (without IBU)	First Day of Observation (with IBU)	Observation after 7 Days (with IBU)	Observation after 3 Months (with IBU)
TP-IBU-SA	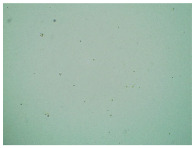	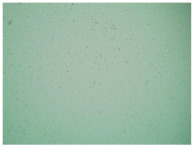	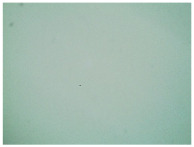	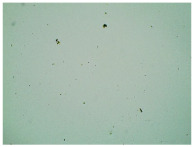
TP-IBU-Menth	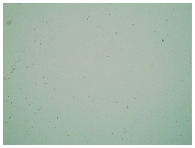	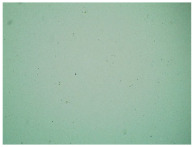	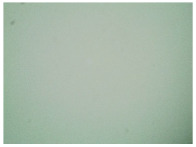	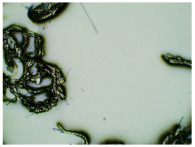
TP-IBU-Urea	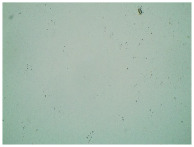		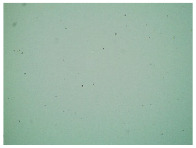	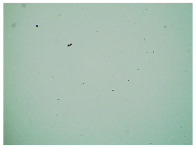
TP-IBU-GA	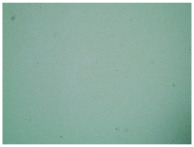	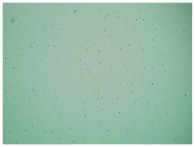	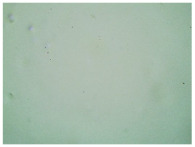	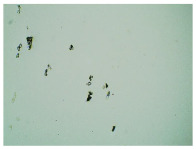
TP-IBU-All	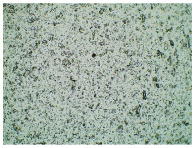	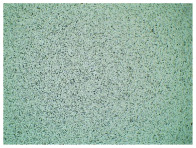	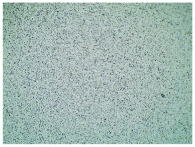	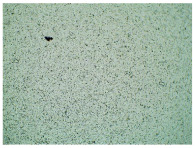
TP-IBU-OA	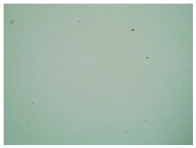	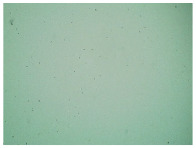	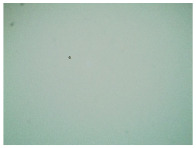	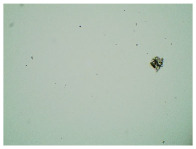
TP-IBU-LA	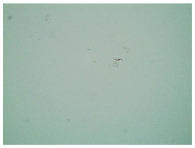	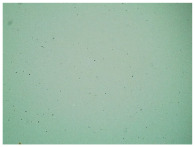	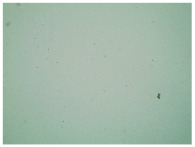	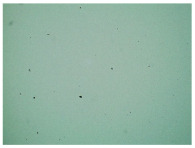
TP-IBU-T80	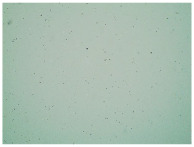	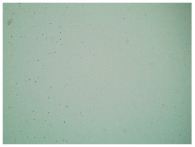	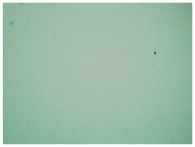	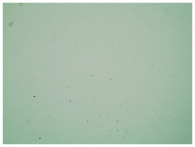
TP-IBU-Cam	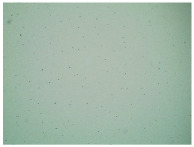	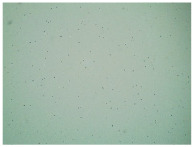	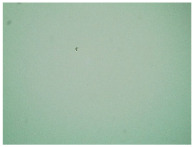	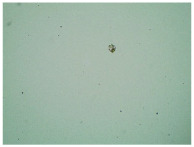
TP-IBU-LC	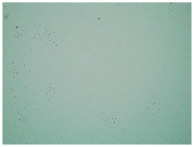	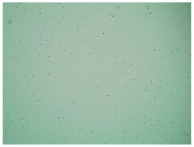	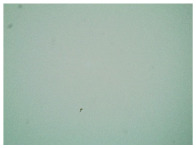	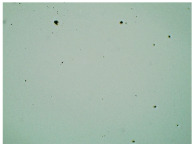
TP-IBU-GL	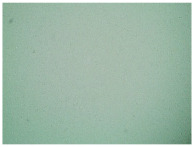	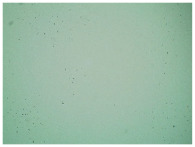	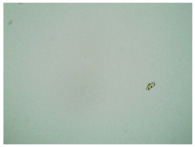	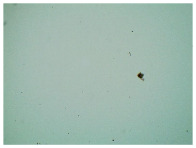
TP-IBU	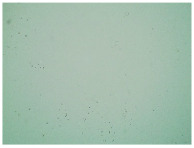	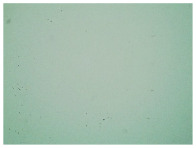	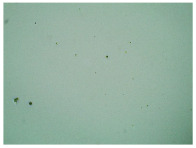	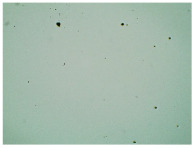

**Table 4 ijms-24-15632-t004:** Microscopic observation of patch samples containing ibuprofen and various permeation promoters during the seasoning time (patches not protected with siliconized foil).

Sample Code	Observation after 7 Days (with IBU) Patches not Protected with Siliconized Foil	Observation after 3 Months (with IBU) Patches not Protected with Siliconized Foil
TP-IBU-Menth	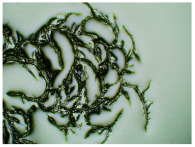	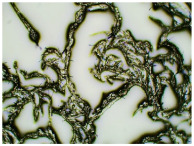
TP-IBU-All	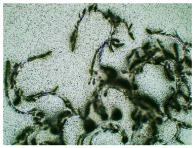	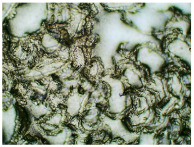
TP-IBU-OA	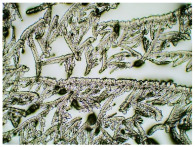	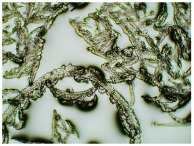
TP-IBU-LC	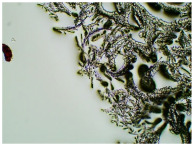	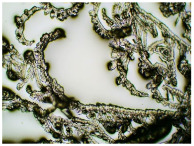
TP-IBU-GL	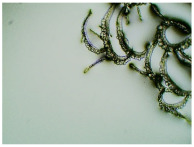	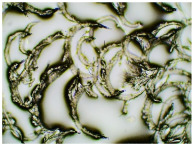

**Table 5 ijms-24-15632-t005:** Thermal properties of obtained transdermal patches.

Sample Code	T_IDT_ (°C)	T_d_^50%^ (°C)	T_MDT_ (°C)	T_g_ (°C)
TP	318.1	360.0	365.3	−45.98 [33]
TP-IBU	172.5	350.4	381.8	−51.88 [33]
TP-IBU-SA	159.0	352.0	373.7	−52.20
TP-IBU-Menth	163.6	336.9	369.7	−51.77
TP-IBU-Urea	167.6	355.0	380.5	−51.15
TP-IBU-GA	164.8	336.3	360.8	−52.09
TP-IBU-All	178.0	351.3	383.1	−52.48
TP-IBU-OA	173.7	344.2	376.3	−55.68
TP-IBU-LA	164.9	351.9	370.1	−50.62
TP-IBU-T80	170.1	358.1	372.9	−53.56
TP-IBU-Cam	166.4	347.1	357.1	−51.54
TP-IBU-LC	174.0	337.4	331.9	−53.81
TP-IBU-GL	161.0	344.3	370.4	−52.17

T_IDT_—onset decomposition temperature, T_d_^50%^—50% weight loss temperature, T_MDT_—maximum decomposition temperature, T_g_—glass transitions.

**Table 6 ijms-24-15632-t006:** Skin permeation parameters of IBU after 24 h of permeation through pig skin.

Sample Code	Cumulative Permeation Mass [µg/cm^2^]	J_SS_ [µg/cm^2^∙h]	K_P_∙10^3^ [cm/h]	Q%_24 h_
TP-IBU	68.386 ± 1.210	7.66 ± 0.24	5.36 ± 0.19	4.79 ± 0.27
TP-IBU-SA	87.259 ± 1.425 *	10.03 ± 0.28	7.02 ± 0.20	6.11 ± 0.10
TP-IBU-Menth	119.463 ± 4.413 *	13.13 ± 0.27	9.19 ± 0.19	8.36 ± 0.31
TP-IBU-Urea	129.762 ± 8.961 *	15.65 ± 0.36	10.96 ± 0.27	9.08 ± 0.22
TP-IBU-GA	150.278 ± 6.226 *	16.00 ± 0.27	11.20 ± 0.37	10.05 ± 0.44
TP-IBU-All	157.861 ± 9.423 *	21.27 ± 0.48	14.89 ± 0.38	11.05 ± 0.21
TP-IBU-OA	163.306 ± 24.418 *	11.47 ± 0.28	8.03 ± 0.25	11.43 ± 0.37
TP-IBU-LA	89.856 ± 2.466 *	6.94 ± 0.24	4.86 ± 0.27	6.29 ± 0.44
TP-IBU-T80	122.473 ± 0.120 *	14.15 ± 0.49	9.90 ± 0.54	8.57 ± 0.35
TP-IBU-Cam	113.804 ± 16.655 *	14.42 ± 0.43	10.08 ± 0.70	7.96 ± 0.40
TP-IBU-LC	144.114 ± 0.524 *	6.73 ± 0.29	4.71 ± 0.20	10.09 ± 0.28
TP-IBU-GL	126.337 ± 13.193 *	6.97 ± 0.44	4.87 ± 0.43	8.84 ± 0.35
CP	83.705 ± 7.184	8.79 ± 0.31	6.15 ± 0.22	5.96 ± 0.37

J_SS_—steady-state flux; K_P_—permeability coefficient; Q_%24 h_—per cent drug permeated after 24 h. *—significant differences of the derivatives compared to the control (TP without enhancer—TP-IBU), *p* < 0.001, α = 0.05, mean ± SD, *n* = 3. The statistically significant difference was estimated by ANOVA using the Tukey’s test.

**Table 7 ijms-24-15632-t007:** Self-adhesive properties of the transdermal patch containing various enhancers.

	Adhesive		Permeation Promoter		Solvent		Ibuprofen ^(1)^	
Sample Code	Symbol	Weight [g]	Symbol	Weight [g]	Symbol	m_1_	weight [g]	m_2_
TP	DT54	9.8	-	0.2	-	-	-	-
TP-IBU			-		-	-	-	-
TP-SA			SA		OE	1:9	-	-
TP-Menth			Menth		OE	1:6	-	-
TP-Urea			Urea		EtOH	1:9	-	-
TP-GA			GA		Ac	1:5	-	-
TP-All			All		EtOH	1:9	-	-
TP-OA			OA		-	-	-	-
TP-LA			LA		-	-	-	-
TP-T80			T80		-	-	-	-
TP-Cam			Cam		OE	1:2	-	-
TP-LC			LC		-	-	-	-
TP-GL			GL		-	-	-	-
TP-IBU-SA	DT54	9.02	SA	0.18	OE	1:9	2	1:1
TP-IBU-Menth			Menth		OE	1:6		
TP-IBU-Urea			Urea		EtOH	1:9		
TP-IBU-GA			GA		Ac	1:5		
TP-IBU-All			All		EtOH	1:9		
TP-IBU-OA			OA		-	-		
TP-IBU-LA			LA		-	-		
TP-IBU-T80			T80		-	-		
TP-IBU-Cam			Cam		OE	1:2		
TP-IBU-LC			LC		-	-		
TP-IBU-GL			GL		-	-		

m_1_—mass ratio of permeation promoter to solvent; m_2_—mass ratio of ibuprofen to solvent (OE); OE—ethyl acetate; EtOH—ethanol; Ac—acetone; ^(1)^ assumption: 200 mg of active substance/140 cm^2^ of transdermal patch; TP—transdermal patches (containing adhesives); TP-IBU—TP with ibuprofen; TP-SA—TP with salicylic acid; TP-Menth—TP with menthol; TP-Urea—TP with urea; TP-GA—TP with glycolic acid; TP -All—TP with allantoin; TP -OA—TP with oleic acid; TP -LA—TP with linoleic acid; TP-T80—TP with Tween 80; TP-Cam—TP with camphor; TP-LC—TP with N-dodecylcaprolactam; TP-GL—TP with glycerine; TP-IBU-SA—TP with IBU and salicylic acid; TP-IBU-Menth—TP with IBU and menthol; TP-IBU-Urea—TP with IBU and urea; TP-IBU-GA—TP with IBU and glycolic acid; TP-IBU-All—TP with IBU and allantoin; TP-IBU-OA—TP with IBU and oleic acid; TP-IBU-LA—TP with IBU and linoleic acid; TP-IBU-T80—TP with IBU and Tween 80; TP-IBU-Cam—TP with IBU and camphor; TP-IBU-LC—TP with IBU and N-dodecylcaprolactam; TP-IBU-GL—TP with IBU and glycerine.

## Data Availability

Most of the data are provided in this work and in Appendix A. Other data that support the findings of this study are available from the corresponding author upon reasonable request.

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
