# Peer review of "Enhancing Transdermal Delivery: Investigating the Impact of Permeation Promoters on Ibuprofen Release and Penetration from Medical Patches—In Vitro Research"

_ijms, 2023, doi:10.3390/ijms242115632_

Round 1

Reviewer 1 Report

Comments

Title:

is slightly misleading, it would be better to specify that in vitro or ex-vivo characterization studies have been conducted.

Introduction:

 In general, it could be improved and detailed. In particular: 

      It would be better to explain why the authors choose these particular Penetration Enhancers;

      It would be better to add information about the transdermal patches and what kind of characteristics are necessary to obtain the desired administration.

Materials and Method:

      In line 438, the author said: "The prepared acrylate patches were stored under constant conditions at 20°C with 50% humidity"

The author needs to explain the reason for these conditions. Are there any recent references about it?

      ​​In line 473, the author said: "The acceptor cell contained 8 ml of pH 7.4 PBS". The author should better describe the drug solubility in PBS and the method of quantifying the drug in PBS.

Results:

      The resolution of Table 3 is very low and the writing in the figures is not clear.

      There are two "Tables 3" with the same description: "Table 3. Microscopic observation of patch samples containing ibuprofen and various permeation promoters during seasoning (patches protected with siliconized foil)." Is it a mistake?. 

      In Table 4 and Table 5, the author must choose to always indicate the same significant digits.

      In line 126, the author said: “the amount of solvents used to dissolve the active substances differed, influencing the solid weight content determined via gravimetry.” The author should explain how the solvent is removed because if not completely removed, it could affect the results.

      In figures 3 and 5, the author should indicate, in both, the time in hours or minutes. Furthermore, it would be advisable to specify the release percentage to facilitate reading the result.

Furthermore, I want to underline that referencing is an important part of the research work, it demonstrates the breadth and depth of the research.Very important is the publication year and I suggest that more recent articles could be added in the bibliography section.

Author Response

Dear Editor,
Dear Reviewer,

We would like to inform you that the manuscript, following the reviewer's comments, has been revised. We hope it now meets the journal's requirements and can be considered for publication.

Detailed responses to the reviewer's comments are included in the appendix.

Thank you for your cooperation.

Sincerely,
Paula Ossowicz-Rupniewska

Reviewer 2 Report

The research presented in the manuscript analyzed the impact of various enhancers from self-adhesive transdermal patches on drug penetration and accumulation. The relationships between various penetration promoters, ibuprofen and the adhesive properties of the patches were examined. In the analyzed systems, the best results were obtained for allantoin patches. The authors point out their potential commercial use. The research is interesting and has application features, but it should be expanded to include at least several drugs with different properties. The discussion should take into account both the structure of the drug substance molecule and its properties (including solubility, lipophilicity, etc.), the type of substances constituting penetration promoters (including those currently commercially used) and the types of adhesives.

Author Response

(The authors gave the same response as above.)

Reviewer 3 Report

The paper entitled "Enhancing Transdermal Delivery: Investigating the Impact of Permeation Promoters on Ibuprofen Release and Penetration from Medical Patches" describes how the inclusion of various permeation promoters (salicylic acid, menthol, urea, glycolic acid, allantoin, oleic acid, Tween 80, linoleic acid, camphor, N-dodecyl caprolactam and glycerin) in acrylic pressure-sensitive adhesive-based transdermal patches can influences the release and permeation of ibuprofen through the skin and its accumulation in the skin. The summary is clear and precise. The Introduction Section is well written and easy to read, highlighting the importance of the topic addressed. The manuscript discusses well the experiments but I have the following comments and suggestions:

- As ibuprofen is present during crosslinking reaction (T=1100C) it is important to investigate the influence of this process on the stability of drug (degradation and formation of by-product ibuprofen);

- The drug interact with the adhesive?

- A comparision with other types of patches that contain ibuprofen or with existing patches on the market should be done.

The paper can be accepted for publication after minor revisions.

Author Response

(The authors gave the same response as above.)
